# Comparison of Microcirculatory Perfusion in Obese and Non-Obese Patients Undergoing Cardiac Surgery with Cardiopulmonary Bypass

**DOI:** 10.3390/jcm10030469

**Published:** 2021-01-26

**Authors:** Chantal A. Boly, Margot Venhuizen, Nicole A. M. Dekker, Alexander B. A. Vonk, Christa Boer, Charissa E. van den Brom

**Affiliations:** 1Department of Anesthesiology, Amsterdam UMC, VU University, 1081 HV Amsterdam, The Netherlands; c.boly@amsterdamumc.nl (C.A.B.); margotveerle@gmail.com (M.V.); n.dekker@amsterdamumc.nl (N.A.M.D.); c.boer@amsterdamumc.nl (C.B.); 2Departments Physiology and Cardiothoracic Surgery, Amsterdam UMC, VU University, 1081 HV Amsterdam, The Netherlands; 3Department of Cardiothoracic Surgery, Amsterdam UMC, VU University, 1081 HV Amsterdam, The Netherlands; aba.vonk@amsterdamumc.nl; 4Faculty of Medicine, Amsterdam UMC, VU University, 1081 BT Amsterdam, The Netherlands; 5Department of Intensive Care, Amsterdam UMC, University of Amsterdam, 1105 AZ Amsterdam, The Netherlands

**Keywords:** cardiac surgery, cardiopulmonary bypass, microcirculation, obesity, SDF, perfusion

## Abstract

Obesity is a frequent comorbidity among patients undergoing cardiac surgery with cardiopulmonary bypass (CPB). Cardiac surgery with CPB impairs microcirculatory perfusion, which is associated with multiple organ failure. As microvascular function is frequently compromised in obese patients, we studied whether cardiac surgery with CPB has a more detrimental effect on microcirculatory perfusion in obese patients. Sublingual microcirculatory perfusion was measured with sidestream dark field (SDF) imaging in obese patients (body mass index ≥32 kg/m^2^; *n* = 14) without type II diabetes mellitus and in lean patients (BMI 20–25 kg/m^2^; *n* = 22) undergoing cardiac surgery with CPB. CPB reduced systolic blood pressure and mean arterial pressure more profoundly in lean compared with obese patients (SBP: 38% vs. 18%; MAP: 11% vs. 8%, *p* < 0.05), and both restored after weaning from CPB. No differences were present in intraoperative glucose, hematocrit, hemoglobin, lactate, and blood gas values between obese and lean patients. Microcirculatory perfusion did not differ between obese and lean patients the day before surgery. CPB decreased microcirculatory perfusion with 9% in both groups, but this was only significant in lean patients (*p* < 0.05). Three days following surgery, microcirculatory perfusion was restored in both groups. In conclusion, microcirculatory perfusion was equally disturbed during cardiac surgery with CPB in metabolically healthy obese patients compared to lean patients.

## 1. Introduction

Obesity is increasing worldwide [1], and, as a result, the proportion of obese patients undergoing cardiac surgery is growing. Obesity during cardiac surgery is a major risk factor for the development of postoperative complications such as myocardial infarction, septicemia [2], and pulmonary and gastrointestinal complications [3] and is also an independent predictor for mortality [4].

Microcirculatory perfusion disturbances are commonly present in patients undergoing cardiac surgery with cardiopulmonary bypass (CPB) [5,6,7,8] and are associated with the development of multiple organ failure [9]. Several perioperative aspects are known to affect microvascular function [10], such as anesthetics [5,11,12], surgical stress, vasoactive drugs, and fluid therapy [10]. In particular, the use of CPB is shown to deteriorate microcirculatory perfusion, as we have previously shown that off-pump cardiac surgery did not impair microcirculatory perfusion [6]. Furthermore, cardiac surgery with CPB induces platelet dysfunction, coagulation activation, hemodilution, inflammation, and endothelial dysfunction [13,14,15,16,17].

The preoperative status of the patient might also affect microcirculatory perfusion during cardiac surgery with CPB. Obese patients are in general at increased risk of complications due to comorbidities, such as a prothrombotic state [18] and microvascular dysfunction [19,20]. In particular, endothelium-dependent capillary recruitment is diminished in obese subjects [21,22,23]. Impaired vasodilation [24] and decreased levels of nitric oxide [19] may further impair vasoreactivity. In contrast, obesity also may play a protective role during cardiac surgery [25,26], which has been described as the “obesity paradox” [27].

Taken together, obesity as preoperative status might affect the risk of developing microcirculatory perfusion disturbances during cardiac surgery with CPB. We hypothesized that cardiac surgery with CPB might have a more detrimental effect on microcirculatory perfusion in obese patients due to compromised microvascular function. We therefore assessed whether obesity additionally affects sublingual microcirculatory perfusion during cardiac surgery with CPB compared to lean patients.

## 2. Materials and Methods

### 2.1. Study Population

This single-center observational clinical study was approved by the local Human Subjects Committee (MeTC VU University Medical Center, approval number 2016.136), and written informed consent was obtained from all participants. Patients were considered eligible for study inclusion when undergoing elective, nonaortic cardiac surgery and if they had a body mass index (BMI) ≥32 kg/m^2^ (obese group) or a BMI <25 kg/m^2^ (lean group). Exclusion criteria were previous cardiac surgery, emergency surgery, and type I and II diabetes mellitus. Patients were screened for the presence of diabetes the day before surgery using a point-of-care device to determine hemoglobin A1c (HbA1c) levels and were excluded when HbA1c was >48 mmol/L or 6.5%. Patients were included for analyses if microcirculation measurements were of good quality on three or more time points.

### 2.2. Anesthesia

Premedication consisted of 5 mg lorazepam on the morning of surgery. For induction of anesthesia, sufentanil (3–7 µg/kg), rocuronium bromide (1 mg/kg), and midazolam (0.1 mg/kg) were used. Anesthesia was maintained by continuous propofol infusion (200–400 mg/h), and all patients received 1 mg/kg dexamethasone and 1000 mg cefazoline. Mechanical ventilation was applied using a 45% O_2_/air mixture, 10 mL/kg tidal volume, and a positive end-expiratory pressure of 5 cm H_2_O.

### 2.3. Cardiopulmonary Bypass

Cardiopulmonary bypass (CPB) was performed according to standard local protocol with an S5 heart–lung machine with a heater–cooler device (Stöckert Instrumente GMBH, Munich, Germany) and centrifugal pump (Sarns, Terumo Europe NV, Leuven, Belgium), combined with a heparin-coated polyvinyl tubing system with a hollow-fiber oxygenator and arterial line filter (Affinity, Medtronic, Minneapolis, MN, USA) as previously described [6,7]. Priming of the circuit was performed with 1000 mL of modified fluid gelatin (Braun Melsungen AG, Melsungen, Germany), 250 mL of lactated Ringer’s solution (Baxter BV, Utrecht, The Netherlands), 100 mL of mannitol (20%; Baxter BV, Utrecht, The Netherlands), and 50 mL of sodium bicarbonate (8.4%; Braun Melsungen AG, Melsungen, Germany), 1000 mg of cefalozine (Eli Lilly Nederland BV, Nieuwegein, The Netherlands), and 5000 IU of porcine heparin. Nonpulsatile CPB (34 °C; 2.2–3.0 L/min/m^2^) started after heparin administration (300 IU/kg) when the activated coagulation time (ACT) exceeded 480 s. Cardiac arrest was induced by cold crystalloid cardioplegia solution (4 °C; St. Thomas, VU University Medical Center, Amsterdam, The Netherlands). Patients were weaned from CPB when rectal temperature was above 36 °C.

### 2.4. Microcirculatory Perfusion Measurements and Analysis

Sublingual microcirculatory perfusion measurements were performed using sidestream dark field (SDF) imaging (Microscan; Microvision Medical, Amsterdam, The Netherlands) as described previously [6,7]. Sublingual microcirculatory perfusion was measured the day before surgery (pre-op) and intraoperatively after induction of anesthesia (anesthesia), 10 min after start of CPB (start CPB) and 10 min after weaning from CPB (post-CPB). Three days after surgery, a final measurement was performed (72 h post-op). Three recordings with 10 s of stable footage were made per time point, and pressure artifacts were excluded. Offline analyses were performed using automatic vascular analysis software (AVA 3.2; Microvision Medical, Amsterdam, The Netherlands), and measurements were averaged for each time point. Blood vessels <20 µm were manually identified and scored for total vessel density (TVD), perfused vessel density (PVD), proportion of perfused vessels (PPV), and microvascular flow index (MFI) according to general consensus [28]. For determination of MFI, each quadrant was scored for flow: no flow (0), sluggish flow (1), intermittent flow (2), or continuous flow (3), and scores were averaged per video. All perfusion variables were normalized to baseline values per patient to minimalize variance caused by different measuring sites.

### 2.5. Hemodynamic and Laboratory Parameters

Systolic, diastolic, and mean arterial pressure, hematocrit, hemoglobin, temperature, and fluid administration were determined intraoperatively after induction of anesthesia (anesthesia), 10 min after start of CPB (start CPB) and 10 min after weaning from CPB (post-CPB). Lactate levels, pH, pCO_2_, base excess, and HCO_3_^−^ in arterial blood gas were determined during surgery at similar time points. Additionally, CPB-related parameters including aortic cross-clamping time, bypass time, and the duration of surgery were recorded.

### 2.6. Statistical Analysis

The primary endpoint of the study was the difference in PVD between groups before surgery and after weaning from CPB. We hypothesized that microcirculatory perfusion was impaired in obese patients at baseline and that the effect of cardiac surgery with CPB further decreased PVD. We previously showed that the change in PVD following CPB is approximately 4 mm/mm^2^ compared to baseline values using the SDF camera in lean subjects [7], and expected a larger decrease in PVD after CPB in obese subjects (an estimated 6 mm/mm^2^. Using a 2-sided test, a standard deviation of 2 mm/mm^2^, an alpha of 0.05, and a beta of 0.8, the estimated sample size was calculated as 16 patients per group.

Data were analyzed using SPSS statistical software package (17.0; IBM, New York, NY, USA). TVD, PVD, and PPV values were normalized to baseline values per patient. All values are expressed as mean ± standard deviation or median with interquartile range (IQR). Differences between groups at baseline were analyzed by a Student’s *t*-test for parametric data or Mann–Whitney U tests for nonparametric data. Time-dependent differences were analyzed with two-way analysis of variance (ANOVA) and Bonferroni post hoc analysis. A *p*-value < 0.05 was considered as statistically significant.

## 3. Results

### 3.1. Patient Characteristics

Patients were included between June 2016 and March 2018. A total of 22 lean and 14 obese patients were included for analysis of microcirculatory function, and their characteristics are shown in Table 1. All patients were ASA 3 (American Society of Anesthesiologists Physical Status), except for one patient in the obese group who was classified as ASA 4. EuroSCORE II was comparable between obese and lean patients. Both groups consisted mainly of male patients; 82% in the lean group and 86% in the obese group. There were no differences in age and length between the groups, and bodyweight and BMI were significantly higher in the obese group compared with the lean group according to the inclusion criteria. The obese group consisted of more patients with preoperative hypertension (64%) compared to the lean group (35%). Obese patients received more often antihypertensive drugs (86%) compared to lean patients (59%). Furthermore, obese patients had higher HbA1c values compared to lean patients, however, this did not reach statistical significance (*p* = 0.09).

### 3.2. Intraoperative Hemodynamics, Temperature, and Lactate Levels

Intraoperative systolic blood pressure (SBP), diastolic blood pressure (DBP), and mean arterial blood pressure (MAP) are shown in Figure 1. Directly following induction of anesthesia, there were no differences in SBP, DBP, and MAP between obese and lean patients. After start of CPB, a significant drop was seen in SBP and MAP, without alterations in DBP. This drop in SBP and MAP was more severe in lean patients compared to obese patients (Figure 1A,C). SBP and MAP fully restored to pre-CPB values after weaning from CPB in both groups.

Intraoperative glucose levels were not affected by the start of CPB, however, glucose levels were significantly increased following weaning from CPB (Figure 2A). This increase was comparable in lean and obese patients. Hematocrit levels significantly decreased following start of CPB in lean as well as obese patients and remained unaltered following weaning from CPB (Figure 2B). Lactate levels were significantly higher in obese patients, compared to lean patients, after induction of anesthesia, however, levels did not significantly change over time in both groups (Figure 2C).

Hemoglobin levels significantly decreased following the start of CPB in lean as well as obese patients and remained unaltered following weaning from CPB (Table 2). pH and pCO_2_ were not affected by CPB, whereas base excess and HCO_3_^−^ significantly decreased over time. No differences were found in pH, pCO_2_, base excess, and HCO_3_^−^ between lean and obese patients (Table 2). Temperature decreased significantly after initiation of CPB in a similar fashion in both groups, and restored in both groups after weaning from CPB.

Aorta cross-clamp time, bypass time, and surgery time did not differ between groups (Table 3). Most patients received noradrenaline or dopamine, in which, compared to obese patients, lean patients received dopamine more often. Compared to obese patients, lean patients seemed to receive more fluids in total, but this did not reach statistical significance (*p* = 0.08). Only one patient from the lean group had preoperative atrial fibrillation, and the development of postoperative atrial fibrillation was similar in both groups (14%). Additionally, there were no differences in intensive care unit (ICU) and hospital stay and in-hospital mortality between lean and obese patients.

### 3.3. Preoperative Microcirculatory Perfusion

Preoperative microcirculatory perfusion did not differ between obese and lean patients, as shown by the perfused vessel density, total vessel density, proportion of perfused vessels, and microvascular flow index (Figure 3).

### 3.4. Intra- and Postoperative Microcirculatory Perfusion

Perfused vessel density (Figure 4A) and total vessel density (Figure 4C) did not change over time nor differ between obese and lean patients. The proportion of perfused vessels decreased directly following weaning from CPB, compared to the proportion of perfused vessels at anesthesia induction, in lean patients (Figure 4B), however, there was no significant decrease in obese patients. The microvascular flow index also decreased in lean patients directly following weaning from CPB compared to anesthesia induction (Figure 4D). This reduction in proportion of perfused vessels and microvascular flow index in lean patients following CPB restored within 72 h following surgery (Figure 4B,D).

## 4. Discussion

In this study, we have demonstrated that sublingual microcirculatory perfusion did not differ between obese patients with a body mass index ≥32 kg/m^2^ without type II diabetes mellitus and lean patients during resting conditions the day before surgery. Following cardiac surgery with cardiopulmonary bypass (CPB), sublingual microcirculatory perfusion was equally disturbed in obese and lean patients independent of intraoperative blood pressure. Three days following cardiac surgery with CPB, sublingual microcirculatory perfusion was restored in both groups. Future studies should focus on the relation between microcirculatory perfusion and postoperative complications following cardiac surgery with CPB in obese and lean patients.

Microvascular function is frequently compromised in obese patients. In the present study, we showed that sublingual microcirculatory perfusion did not differ between obese and lean patients during resting conditions determined at the day before surgery. Although measured in different vascular beds, this finding is in agreement with previous studies showing no differences in skin perfusion [29,30] and myocardial perfusion [31] during resting conditions between obese and lean patients. These findings might be due to the fact that microvascular dysfunction in obese patients is commonly characterized by a reduced ability to vasodilate and an enhanced vasoconstrictor response [29,30], and suggest that differences will manifest during stress conditions, such as cardiac surgery with CPB. Taken together, the absence of sublingual microcirculatory perfusion disturbances in obese compared to lean patients might be due to nonstressed, resting conditions.

Microcirculatory perfusion disturbances in patients undergoing cardiac surgery with CPB were previously shown by our group and others [5,6,7,8]. Unexpectedly, we did not find a more detrimental effect of cardiac surgery with CPB on sublingual microcirculatory perfusion in obese compared to lean patients. This might be due to factors such as the vascular bed studied, technique used, gender, or medication. Our study population consisted mainly of men, while previous studies predominantly found alterations in obese women [29,32,33]. Additionally, drugs such as aspirin, antihypertensive agents, and statins are known to affect tissue perfusion [34,35] and might have blunted the effect of cardiac surgery with CPB on microcirculatory perfusion in obese patients. In addition, intraoperative factors such as anesthetics have previously shown to influence microvascular function and tissue perfusion [5,11,12,36]. However, anesthesia induction did not affect sublingual microcirculatory perfusion in obese nor in lean patients in the present study. In addition, CPB itself has negative effects on microcirculatory perfusion, as off-pump cardiac surgery did not impair microcirculatory perfusion [6]. This effect is partly explained by hemodilution, but also hypothermia, inflammation, and alterations in blood pressure play a role [13,37]. In the present study, no differences were seen in intraoperative hematocrit levels as reflection of hemodilution or body temperature between obese and lean patients. We did find differences in intraoperative blood pressure between obese and lean patients. These differences might be due to a higher pump flow, as the pump flow is based on body surface area (BSA). However, the differences in intraoperative blood pressure did not translate into differences in microcirculatory perfusion between obese and lean patients. This suggests that microcirculatory perfusion changes occurred independently of changes in blood pressure, which is in agreement with the previously described loss of hemodynamic coherence [38]. Interestingly, the anti-inflammatory effects of adipose tissue have been suggested as a possible protective mechanistic explanation in obesity [39]. In addition, CPB-induced inflammation and acute kidney injury were reduced in obese swines [39]. As inflammatory markers were not measured, we can only speculate that changes in CPB-induced inflammatory response might explain the absence of a more detrimental effect on microcirculatory perfusion in obese patients. Taken together, sublingual microcirculatory perfusion was equally disturbed in obese and lean patients independent of intraoperative blood pressure.

Certain limitations of this study need to be mentioned. This study was limited by the incomplete recruitment of obese patients and the relatively small group size of obese patients. Inclusion was hindered as obese patients frequently presented with (pre)diabetes. At the same time, one of the strengths is that we included metabolically healthy obese patients, as HbA1c was measured to exclude patients with type II diabetes mellitus. Another limitation might be that inclusion was based on BMI and not on body surface area (BSA). BMI as measure of adiposity is limited as it cannot identify differences in body composition and body fat [40], and visceral fat and subcutaneous fat differently express inflammatory genes [41].

## 5. Conclusions

In conclusion, we showed that sublingual microcirculatory perfusion did not differ between obese patients with a body mass index ≥32 kg/m^2^ without type II diabetes mellitus and lean patients during resting conditions the day before surgery. Additionally, sublingual microcirculatory perfusion was equally disturbed in obese and lean patients during cardiac surgery with cardiopulmonary bypass, independent of the differences in intraoperative blood pressure. Three days following cardiac surgery with cardiopulmonary bypass, sublingual microcirculatory perfusion was restored in both groups. These data report that obesity has a neutral effect on microcirculatory perfusion in response to cardiac surgery with cardiopulmonary bypass, suggesting that cardiac surgery should not be withheld from obese patients based on the microcirculatory response in comparison with their lean counterparts. However, future studies need to focus on the relation between microcirculatory perfusion and postoperative complications following cardiac surgery with cardiopulmonary bypass.

## Figures and Tables

**Figure 1 jcm-10-00469-f001:**
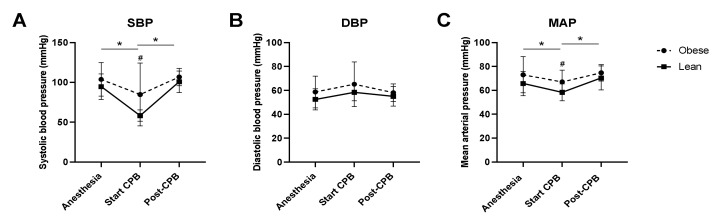
Intraoperative blood pressure. Mean arterial blood pressure (MAP; (**A**)), systolic blood pressure (SBP; (**B**)), and diastolic blood pressure (DBP; (**C**)) measured after induction of anesthesia (anesthesia), 10 min after start of cardiopulmonary bypass (start CPB), and 10 min after weaning from CPB (post-CPB) in lean (black line) and obese (dotted line) patients. Data are represented by mean ± SD and tested with a two-way ANOVA and Bonferroni post hoc analysis; * *p* < 0.05 CPB effect; # *p* < 0.05 obese vs. lean patients.

**Figure 2 jcm-10-00469-f002:**
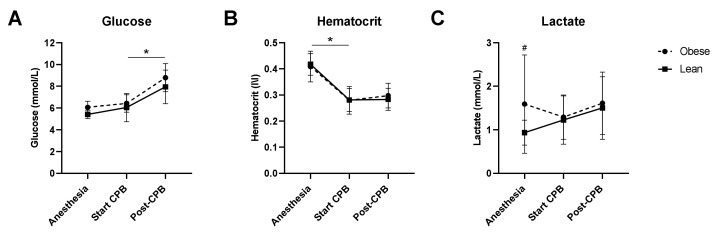
Intraoperative glucose, hematocrit, and lactate levels. Glucose (**A**), hematocrit (**B**), and lactate (**C**) measured after induction of anesthesia (anesthesia), 10 min after start of cardiopulmonary bypass (start CPB), and 10 min after weaning from CPB (post-CPB) in lean (black line) and obese (dotted line) patients. Data are represented by mean ± SD and tested with a two-way ANOVA and Bonferroni post hoc analysis; * *p* < 0.05 time effect; # *p* < 0.05 obese vs. lean patients.

**Figure 3 jcm-10-00469-f003:**
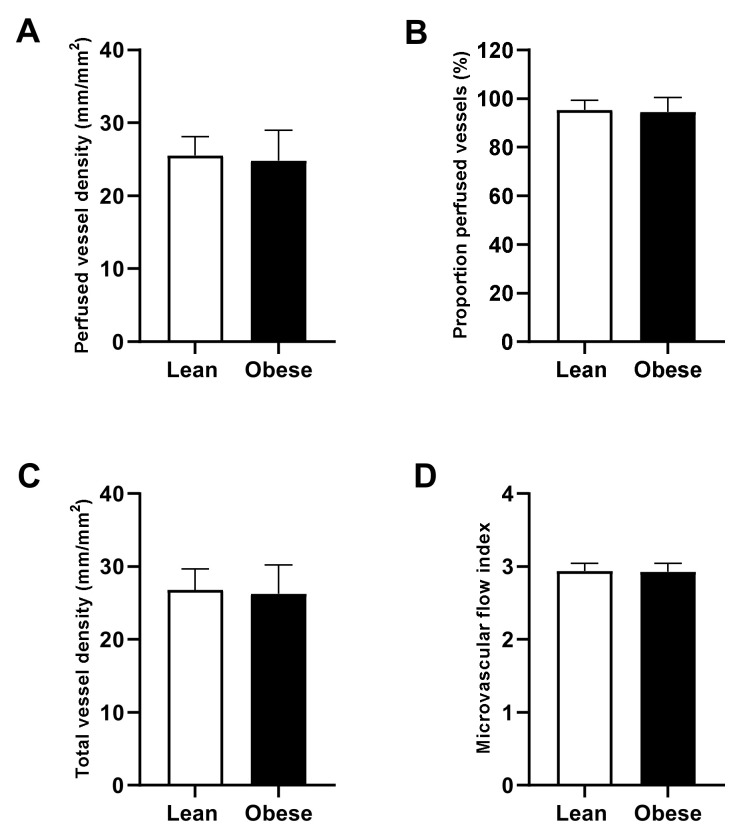
Preoperative microcirculatory perfusion. Perfused vessel density (**A**), total vessel density (**B**), proportion of perfused vessels (**C**), and microvascular flow index (**D**) measured one day before surgery in lean (open bars) and obese (black bars) patients. Data are represented by mean ± SD and tested with a student *t*-test.

**Figure 4 jcm-10-00469-f004:**
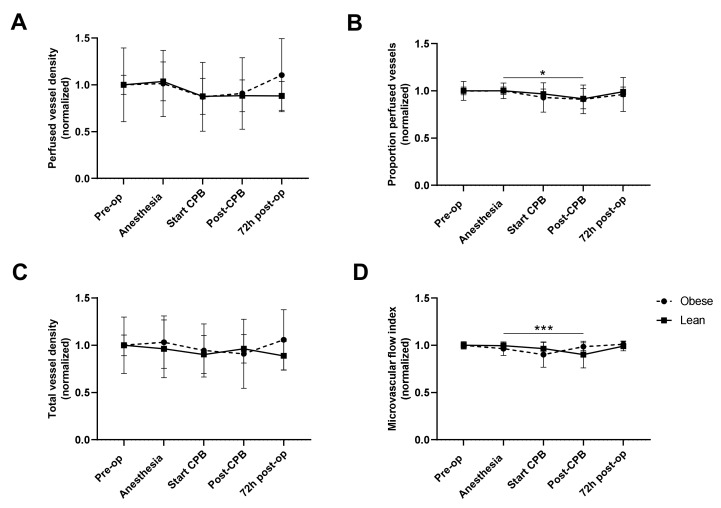
Sublingual microcirculatory perfusion as shown by perfused vessel density (**A**), proportion of perfused vessels (**B**), total vessel density (**C**), and microvascular flow index (**D**) measured before surgery (pre-op), after induction of anesthesia (anesthesia), 10 min after start of cardiopulmonary bypass (start CPB), 10 min after weaning from CPB (post-CPB), and 72 h after end of surgery (72 h post-op) in lean (black line) and obese (dotted line) patients. Data are normalized to pre-op, represented by mean ± SD, and tested with a two-way ANOVA and Bonferroni post hoc analysis; * *p* < 0.05, *** *p* < 0.001 time effect in lean patients.

**Table 1 jcm-10-00469-t001:** Patient characteristics.

	Lean (*n* = 22)	Obese (*n* = 14)	*p*-Value
Gender (male/female)	18/4	12/2	0.99
Age (years)	72 ± 9	67 ± 11	0.14
Length (cm)	177 ± 9	173 ± 7	0.11
Weight (kg)	73 ± 9	104 ± 14	<0.0001
BMI (kg/m^2^)	23.2 ± 1.4	34.8 ± 2.8	<0.0001
Smoker (yes/no)	3/19	1/13	0.99
Preoperative creatinine (µmol/L)	86.2 ± 13.2	98.1 ± 24.9	0.07
Preoperative hemoglobin (mmol/L)	8.8 ± 0.8	8.7 ± 1.0	0.82
Preoperative HbA1c (mmol/L)	39.2 ± 3.7	42.0 ± 4.4	0.09
Hypertension (yes/no)	8/15	9/5	0.10
Antihypertensive medication (yes/no)	9/13	12/2	0.01
Beta-blocker (*n*)	4	4	
ACE inhibitor/ATII antagonist (*n*)	1	2	
Calcium blocker (*n*)	0	1	
Diuretics (*n*)	1	0	
Combination (*n*)	3	5	
Type of surgery (*n*)	CABG: 8	CABG: 9	
	AVR: 8	AVR: 4	
	MVR: 4	MVR: 0	
	AVR/CABG: 2	AVR/CABG: 1	
EuroSCORE II (%)	1.3 (0.7–2.3)	1.5 (1.1–3.3)	0.39

Data are represented by mean ± SD, median (IQR) or by frequencies and tested with a Student *t*-test, Mann–Whitney test or Fisher’s exact test, respectively. BMI = body mass index, HbA1c = hemoglobin A1c, ACE = angiotensin converting enzyme, ATII = Angiotensin II, CABG = coronary artery bypass graft surgery, AVR = aortic valve replacement surgery, MVR = mitral valve replacement, CPB = cardiopulmonary bypass.

**Table 2 jcm-10-00469-t002:** Intraoperative hemoglobin, temperature, and blood gas values.

		Anesthesia	Start CPB	Post-CPB
Hemoglobin (mmol/L)	Lean	8.4 ± 0.8	5.6 ± 0.9 *	5.7 ± 0.9
	Obese	8.3 ± 1.2	5.6 ± 1.1 *	6.0 ± 1.0
pH	Lean	7.4 ± 0.04	7.4 ± 0.04	7.4 ± 0.04
	Obese	7.4 ± 0.02	7.4 ± 0.02	7.4 ± 0.03
pCO_2_ (mmHg)	Lean	5.25 ± 0.62	5.24 ± 0.41	5.19 ± 0.37
	Obese	5.43 ± 0.46	5.31 ± 0.25	5.22 ± 0.44
Base excess (mEq/L)	Lean	0.5 ± 1.8	−0.1 ± 1.7	−1.3 ± 1.7 *
	Obese	1.1 ± 1.4	−0.04 ± 1.6	−1.4 ± 1.3
HCO_3_^−^ (mmol/L)	Lean	24.5 ± 1.9	24.1 ± 1.4	23.0 ± 1.3 *
	Obese	24.9 ± 1.4	24.1 ± 1.6	23.0 ± 1.3
Temperature (°C)	Lean	36.3 ± 0.4	35.0 ± 1.3	36.4 ± 0.3
	Obese	36.4 ± 0.4	35.5 ± 0.6	36.5 ± 0.2

Intraoperative hemoglobin, temperature, and blood gas values of obese and lean patients after induction of anesthesia (anesthesia), 10 min after start of cardiopulmonary bypass (start CPB), and 10 min after weaning from CPB (post-CPB). Data are represented by mean ± SD; * *p* < 0.05 CPB effect.

**Table 3 jcm-10-00469-t003:** Intraoperative parameters and postoperative outcome.

	Lean (*n* = 22)	Obese (*n* = 14)	*p*-Value
Cross-clamp time (min)	86 ± 42	75 ± 18	0.37
Bypass time (min)	118 ± 51	112 ± 23	0.68
Surgery time (min)	265 ± 89	248 ± 44	0.51
Noradrenaline (yes/no)	6/16	5/9	0.72
Dopamine (yes/no)	21/1	10/4	0.06
Nadir hematocrit	0.26 ± 0.03	0.28 ± 0.06	0.47
Nadir hemoglobin (mmol/L)	5.2 ± 0.7	5.5 ± 1.3	0.38
Highest lactate (mmol/L)	2.5 ± 0.8	2.5 ± 0.6	0.96
Nadir pH	7.3 ± 0.04	7.3 ± 0.04	0.83
CPB outflow temperature (°C)	34.4 ± 1.5	34.3 ± 0.7	0.42
Total fluids (mL)	3985 ± 1237	3052 ± 1921	0.08
Atrial fibrillation (yes/no)	3/18	2/12	0.99
ICU stay (days)	1 [1–1]	1 [1–1]	0.27
Hospital stay (days)	7 [5–9.5]	5 [4–8]	0.38
In-hospital mortality (*n*)	1	0	0.99

Data are represented by mean ± SD, median (IQR) or by frequencies and tested with a Student *t*-test, Mann–Whitney test or Fisher’s exact test, respectively. CPB = cardiopulmonary bypass, ICU = intensive care unit.

## Data Availability

The data is not publicly accessible. Readers are welcome to contact authors to obtain access to the raw data.

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
