# Peer review of "Comparison of Microcirculatory Perfusion in Obese and Non-Obese Patients Undergoing Cardiac Surgery with Cardiopulmonary Bypass"

_jcm, 2021, doi:10.3390/jcm10030469_

Round 1

Reviewer 1 Report

The manuscript is intriguing, methodologically sound, and well written. Nevertheless, several issues might deserve a comment. Though a power calculation was performed one of the study group did not reach the appropriate size. There is no mention on the length of study enrollment, and nothing is reported on the features of the derivation sample. It might be important to know how the selected obese group compares with the overall overweight population referred for cardiac surgery. Finally, a clinical bottom line should be implemented to better understand the significance of study results on daily practice.

Author Response

We would like to thank the reviewer for putting his/her time and care in critically reviewing our manuscript and providing valuable comments that have significantly improved the manuscript. We answered the comments below in a point-wise manner. Changes and additions to the manuscript based on the provided comments have been indicated by track changes.

The manuscript is intriguing, methodologically sound, and well written.

We would like to thank you for your kind words.

Nevertheless, several issues might deserve a comment. Though a power calculation was performed one of the study group did not reach the appropriate size. There is no mention on the length of study enrollment, and nothing is reported on the features of the derivation sample. It might be important to know how the selected obese group compares with the overall overweight population referred for cardiac surgery.

Patient inclusion started in June 2016 immediately following approval of our study by the Human Subjects Committee. The last patient was included in March 2018. We have added this to section 3.1 on page 4, line 148.

In our hospital around 1/3 of the patients undergoing cardiac surgery with cardiopulmonary bypass has a normal weight (BMI 20 – 25 kg/m2), whereas 2/3 is overweight (BMI 25 – 30 kg/m2) or even obese BMI >30 kg/m2). We have chosen for a BMI > 32 kg/m2 as we preferred to include obese patients without type II diabetes mellitus, but who might be metabolically challenged during cardiac surgery with cardiopulmonary bypass.

Unfortunately, we had some issues with the camera for sublingual perfusion measurements and this was followed by the corona pandemic. In our hospital, cardiac surgery is still limited to emergency surgeries. We didn’t find differences in sublingual microcirculatory perfusion between obese and lean patients during cardiac surgery and as we do not expect that including 2 more obese patients will change these results, we decided to stop the study and share our results.

Finally, a clinical bottom line should be implemented to better understand the significance of study results on daily practice.

We agree with the reviewer that this part is limited. We extended this paragraph at page 12, lines 321-325 as follows: “These data report that obesity has a neutral effect on microcirculatory perfusion in response to cardiac surgery with cardiopulmonary bypass and suggests that cardiac surgery should not be withheld from obese patients based on the microcirculatory response in comparison with their lean counterparts.“

Reviewer 2 Report

I read with the interest the manuscript on comparison of microcirculation in obese patients going for cardiac surgery.

The authors conducted a descriptive study single center study in on sublingual microcirculation at different steps of cardiac surgery under CPB from baseline at the operating room to Day 3 after surgery.

They compared lean patients (BMI at 20-25) to obese patients (BMI over 32). As reminded by the authors several times, the screened for obese without type 2 diabetes which is approritate 

The method of assessment was the Cytocam with the Microvision Medical software allowing automated measures.

Finally, the concluded that red blood cell velocity as capillary perfusion are worsened in the same extent in both groups

I have some comments to address to the authors

  1. I would change the title of the manuscript with a more factual one "comparison of microcirculation in obese and non obese... in cardiac surgery?... The current title suggests a specific pattern in obese patients
  2. The main criticism concerns the small and non equal sample size in each group. I am aware that obese without diabetes might be difficult to screen.          But to help the readers the authors should be more exhaustive in data reports as follows
  3. Table 1 (Euroscore II or logistic Euroscore, baseline protide, creatinine and hemoglobin; proportion of patients with auricular fibrillation)
  4. during surgery, proportion of patients with norepinephrine, dobutamine, hematocrit, hemoglobin, arterial lactate, PaCO2, pH, CPB outflow temperature (I would add a table 2 with all time points measurements with micro and macrocirculation variables)
  5. postoperative outcomes are lacking to interpret the final measurement at 72h: proportion of vasoplegic shock, proportion of cariogenic shock, patients under IABP, ECLS, hospital and ICU mortality or in hospital mortality
  6. In table 1 please add the P value for comparisons and present proportions are number (percentage)
  7. Why did the authors choose a cut off of 32 to define obesity rather than 30 (WHO definition)?
  8. In table 1 what total fluid correspond to? In ICU stay? ICU+ surgery?
  9. In table 2, regarding hemoglobin value at anesthesia time, why the patients are anemic considering that they did not receive the CPB priming volume yet? Please add the hematocrit as well.

Author Response

We would like to thank the reviewer for putting his/her time and care in critically reviewing our manuscript and providing valuable comments that have significantly improved the manuscript. We answered the comments below in a point-wise manner. Changes and additions to the manuscript based on the provided comments have been indicated by track changes.

I read with the interest the manuscript on comparison of microcirculation in obese patients going for cardiac surgery. The authors conducted a descriptive study single center study in on sublingual microcirculation at different steps of cardiac surgery under CPB from baseline at the operating room to Day 3 after surgery. They compared lean patients (BMI at 20-25) to obese patients (BMI over 32). As reminded by the authors several times, the screened for obese without type 2 diabetes which is appropriate. The method of assessment was the Cytocam with the Microvision Medical software allowing automated measures. Finally, the concluded that red blood cell velocity as capillary perfusion are worsened in the same extent in both groups.

I have some comments to address to the authors

  1. I would change the title of the manuscript with a more factual one "comparison of microcirculation in obese and non-obese... in cardiac surgery?... The current title suggests a specific pattern in obese patients.

We appreciate this nice suggestion for the title and changed it accordingly: “Comparison of microcirculatory perfusion in obese and non-obese patients undergoing cardiac surgery with cardiopulmonary bypass”.

  1. The main criticism concerns the small and non-equal sample size in each group. I am aware that obese without diabetes might be difficult to screen. But to help the readers the authors should be more exhaustive in data reports as follows
    • Table 1 (Euroscore II or logistic Euroscore, baseline protide, creatinine and hemoglobin; proportion of patients with auricular fibrillation) during surgery, proportion of patients with norepinephrine, dobutamine, hematocrit, hemoglobin, arterial lactate, PaCO2, pH, CPB outflow temperature (I would add a table 2 with all time points measurements with micro and macrocirculation variables)

We have extended our preoperative data in table 1 on page 5 with euroSCORE II, preoperative creatinine and preoperative hemoglobin levels as suggested. We also added an extra table (page 8, table 3) to the manuscript in which we extended intraoperative variables. In this table we report on noradrenaline and dopamine administration and nadir hematocrit, nadir hemoglobin, highest lactate, nadir pH and CPB outflow temperature. We thank the reviewer for these suggestions.

  1. Postoperative outcomes are lacking to interpret the final measurement at 72h: proportion of vasoplegic shock, proportion of cariogenic shock, patients under IABP, ECLS, hospital and ICU mortality or in hospital mortality.

We agree with the reviewer that postoperative outcomes are of interest. Due to the small sample size we feel that the interpretability is limited, however, still of interest. We have added pre-existent atrial fibrillation (page 7, lines 215-216) and the development atrial fibrillation postoperatively and in-hospital mortality (page 8, table 3). No differences were found in postoperative atrial fibrillation, ICU and hospital stay and in-hospital mortality.

  1. In table 1 please add the P value for comparisons and present proportions are number (percentage)

We have added the p-values to table 1 at page 5.

  1. Why did the authors choose a cut off of 32 to define obesity rather than 30 (WHO definition)?

We agree with the reviewer that a cut off of a BMI >32 kg/m2 might be unexpected as the WHO definition of obesity is indeed a BMI between 30 and 35 kg/m2. We have chosen to include patients with a BMI > 32 kg/m2 as we preferred to include obese patients without type II diabetes mellitus, but who might be metabolically challenged during cardiac surgery with cardiopulmonary bypass.

  1. In table 1 what total fluid correspond to? In ICU stay? ICU+ surgery?

We apologize for the indistinctness. The fluids reported in table 1 are given during surgery. We have added this comment to the legend of table 3.

7. In table 2, regarding hemoglobin value at anesthesia time, why the patients are anemic considering that they did not receive the CPB priming volume yet? Please add the hematocrit as well.

In table 1 at page 5, we report that obese and lean patients had an average preoperative hemoglobin level of 8.7 and 8.8 mmol/L (14.0 and 14.2 g/dL), respectively. At the start of the surgery following anesthesia induction, hemoglobin levels were 8.3 and 8.4 mmol/L (13.4 and 13.5 g/dL). According to the WHO, these patients are not anemic at the start of the surgery. In both patient groups hemoglobin levels decreased with 33% following start of CPB, which is accompanied by a reduction in hematocrit as shown in figure 2B.

Round 2

Reviewer 1 Report

Please add a statement on incomplete recruitment in study limitations paragraph. 

Author Response

We have added a statement on the incomplete recruitment of obese patients on page 11, line 307.

Reviewer 2 Report

The authors brought clarifications to all comments.

I have no further request.

Author Response

We thank the reviewer for his/her time!